# Schiff Base Compounds as Fluorescent Probes for the Highly Sensitive and Selective Detection of Al^3+^ Ions

**DOI:** 10.3390/molecules28073090

**Published:** 2023-03-30

**Authors:** Yanling Pang, Desu Meng, Jian Liu, Shengxia Duan, Jingru Fan, Longyu Gao, Xinshu Long

**Affiliations:** 1Department of Chemistry and Engineering, Heze University, Heze 274500, China; 2College of Agriculture and Bioengineering, Heze University, Heze 274000, China

**Keywords:** Schiff base, Al^3+^, fluorescent probe, content detection

## Abstract

Two new Schiff base fluorescent probes (L and S) were designed for selectively detecting Al^3+^ ions in aqueous medium. Structural characterization of the purely synthesized compounds was acquired by IR, ^1^H NMR and ^13^C NMR. Moreover, their photochromic and fluorescent behaviors have been investigated systematically by UV–Vis absorption and fluorescence spectra. The two probes have both high selectivity and sensitivity toward Al^3+^ ions in aqueous medium. The 2:1 stoichiometry between the Al^3+^ and probes was verified by Job’s plot. Moreover, the limits of detection (LOD) for Al^3+^ by L and S were 1.98 × 10^−8^ and 4.79 × 10^−8^ mol/L, respectively, which was much lower than most previously reported probes. The possible recognition mechanism was that the metal ions would complex with Schiff base probes because of the prevalence of the species optimal for complex formation, inhibiting the structural isomerization of conjugated double bonds (-C=N-), inhibiting the proton transfer process in the excited state of the molecules and resulting in changes of its color and fluorescence behavior. Furthermore, the probes will have potential applications for selectively, detecting Al^3+^ ions in the environmental system with high accuracy and providing a new strategy for the design and synthesis of multi-functional sensors.

## 1. Introduction

Aluminum, as one of the most abundant metal elements in the Earth’s crust, has been widely applied in day-to-day life in various sectors, such as food additives, water purification, pharmaceuticals, environmental, building construction, clinical drugs, automobiles, and so on [1,2,3]. However, the extensive use of aluminum products in the above industries might lead to high amounts of aluminum ions (Al^3+^) in water, soil and even in the atmosphere, resulting in serious environmental problems [4,5,6]. The excess accumulation of aluminum ion in the human body may lead to serious health problems, such as Alzheimer’s, Parkinson’s, Menkes, and Wilson’s diseases [7,8,9]. Moreover, the daily intake of aluminum is specified to be 3–10 mg/day per kg body mass, and the limits of Al^3+^ concentration in drinking water is 7.41 mM, according to the World Health Organization’s assessment [10]. Therefore, it is of great practical significance to study the efficient and sensitive detection method for detecting Al^3+^ ions in consideration of environmental protection and human health.

At present, there are some reported techniques for detecting Al^3+^ ions, such as inductively coupled plasma atomic emission spectrometry, atomic absorption spectroscopy, inductively coupled plasma mass spectrometry, selective electrode method, and voltammetry [11,12,13]. However, these technologies require expensive instruments and time-consuming procedures, which increase the workload of inspection to a certain extent. Therefore, it is necessary to develop a direct and simple method for detecting Al^3+^ ions with excellent selectivity and high sensitivity. In recent years, the fluorescent probe method has been paid great attention because of its good selectivity, rapid detection, high sensitivity, simple synthesis and convenient operation [14,15,16,17,18]. As a result, various types of fluorescent probes have been designed and prepared to detect Al^3+^ ions, including hydrazone compounds [19], rhodamine B [20], quinolinecarboxamide-coumarin [21], naphthalimide (CDs) [22], quinoline [23], and Schiff base compounds [24,25,26]. Among these fluorescent probes, Schiff base compounds have gained great attention because of their huge number of positives, such as convenient synthesis, adjustable electronic properties, and excellent chelating ability [27]. For instance, Kaya et al. synthesized a carbazole-based Schiff base chemosensor through one-pot synthesis using 2-hydroxy-1-naphtaldehyde for fluorescent sensing of Al^3+^ ions, and the LOD was found to be 2.59 × 10^−7^ M [25]. Das and co-workers also prepared a simple Schiff base as an effective fluorescent sensor for Al^3+^ ions, and this Schiff base could achieve selective detection of Al^3+^ over other metal ions, such as Zn^2+^, Hg^2+^, Cd^2+^, Pb^2+^, Mn^2+^ and so on [28]. Qi et al. also synthesized two Schiff base fluorescent-colorimetric probes for selectively detecting Al^3+^ ions [29].

Furthermore, Schiff bases have some outstanding advantages, such as higher yield, more stable structure, strong photo-physical properties and stronger fluorescence activity. Schiff base fluorescence probes, owing to hard-base donor sites with nitrogen–oxygen-rich coordination environments, not only have a more stable structure but also have properties of naked-eye recognition and fluorescence detection. Moreover, the conjugated double bonds (-C=N-) in Schiff base fluorescent probes can obtain structural isomerization, which would accelerate the non-radiative transition of the Schiff base in the excited state, further leading to the weak fluorescence of the Schiff base. After adding metal ions, the metal ions would complex with the Schiff base probes, and the protons of the donor will be removed, thereby preventing the structural isomerization of conjugated double bonds (-C=N-), inhibiting the proton transfer process in the excited state of the molecules and resulting in changes of its color and fluorescence behavior. Herein, two Schiff base fluorescent probes were successfully prepared through the one-step method. Synthesized fluorescent probes L and S can effectively recognize Al^3+^ in the DMSO/H_2_O mixed system with advantages of short recognition time, low cost and low detection limit. The synthesized procedure of probes L and S are shown in Figure 1. Moreover, a potential light-induced electron transfer mechanism was proposed according to Job’s analysis. Thus, it is promising to prepare new chemo-sensors for detecting Al^3+^ ions by using Schiff bases.

## 2. Results and Discussion

### 2.1. Fluorescence Emission Spectral Responses of Probes L and S

The fluorescence emission spectra of solutions for probes L and S (5.0 × 10^−5^ mol/L) were investigated at room temperature with the following several kinds of metal ions (2.5 × 10^−4^ mol/L), including Al^3+^, Mg^2+^, Pb^2+^, Zn^2+^, Cu^2+^, Mn^2+^, Co^2+^, Cr^3+^, Hg^2+^, Cd^2+^, Ni^2+^ and Fe^3+^ in DMSO/H_2_O (*v*/*v* = 8:2), as shown in Figure 1a,b. It could be observed that the additions of Al^3+^ ions to the solutions containing probes L/S brought about the high fluorescence emission peak at 516 and 518 nm, which can be ascribed to coordination of probes to Al^3+^, further increasing its structure rigidity and inhibiting the proton transfer process in the excited state of the molecules and C=N isomerization process [30]. Additionally, the fluorescence response of L/S to other metal ions was almost neglected, which was similar to that of the blank probe, indicating that L/S was a specific “turn-on” probe toward Al^3+^ ions with high sensitivity. Furthermore, the color of the L/S solution changed from colorless to bright cyan fluorescence after adding Al^3+^ ions under irradiation of a 430/420 nm lamp, while other metal ions brought about no naked-eye fluorescence, further demonstrating the high selectivity of probes [31]. Furthermore, UV–Vis absorption spectra of probes L and S were also illustrated in Figure 1c,d. The aqueous solution of free L and S exhibited the absorption band located at 345 and 350 nm, respectively. However, two new absorption peaks were presented at 302, 430 and 303, 420 nm for probes L and S, respectively, with the disappearing of absorption peaks at 345 and 350 nm after the addition of Al^3+^ ions. Hence, it could be concluded that both probes L and S possessed excellent sensitivity and complexation ability for detecting Al^3+^ ions.

To further confirm the selectivity of fluorescent probes L and S for detecting Al^3+^ ions, investigations on the influence of general coexistence ions were also carried out by adding Al^3+^ ions (2.5 × 10^−4^ mol/L) into probe L and S (5.0 × 10^−5^ mol/L) solutions containing Mg^2+^, Pb^2+^, Zn^2+^, Cu^2+^, Mn^2+^, Co^2+^, Cr^3+^, Hg^2+^, Cd^2+^, Ni^2+^ and Fe^3+^ (2.5 × 10^−4^ mol/L), separately, shown in Figure 2. It can be seen that the other measured metal ions made no great difference to the fluorescence intensity and UV absorbance of probes L (Figure 2a,c) and S (Figure 2b,d) with Al^3+^, suggesting that the other coexistent metal ions have little effect on probes L and S as good fluorescent probes for the selective detection of Al^3+^ ions.

### 2.2. Quantitative Identification of Al^3+^ by Probes L and S

In order to investigate the effect of Al^3+^ addition on the fluorescence and ultraviolet absorption spectra of probes L and S, fluorescence and ultraviolet absorption experiments were carried out in DMSO/H_2_O (*v*:*v* = 8:2) solution. As illustrated in Figure 3a,b, fluorescence intensities of L and S gradually increased with increasing Al^3+^ concentration (0–150 μM). Moreover, the fluorescence intensity of both probes L and S had a good linear relation between fluorescence intensity and the Al^3+^ concentrations from 0 to 150 μM, as shown in Figure 3c,d. This fitting result clearly indicated that both probes L and S can be realized in fluorescence quantitative detection of trace Al^3+^ ions. Furthermore, detection limits of L and S were calculated to be 1.98 × 10^−8^ and 4.79 × 10^−8^ mol/L, respectively, by the formula of LOD = 3*σ*/*K*, according to the fluorescence intensity results, which were much lower than most previously reported probes for Al^3+^ detection [32,33,34,35,36,37,38,39,40], as shown in Table 1. This result clearly demonstrated that both probes L and S could be applied as fluorescent probes for detecting Al^3+^ ions with high sensitivity and specificity.

Additionally, the UV spectra of both L and S were illustrated in Figure 4a,b with the addition of various Al^3+^ ions concentration. The absorption peak of probe L at 345 nm gradually disappeared with the increase in Al^3+^ concentration, and two new absorption peaks appeared at 430 and 302 nm, and absorption intensity at 430 and 302 nm constantly increased with the increase in Al^3+^ concentration, accompanied with no position change of the peak. The absorbance ratio (y) of probe L at 430 and 302 nm has a good linear relationship with the concentration of Al^3+^ in the range of 1.0 × 10^−6^~5.5 × 10^−5^ mol/L with the detection limit of 3.65 × 10^−8^ mol/L, as shown in Figure 4c. For probe S, the UV absorption peak experienced similar changes in comparison to that of probe L. The absorption peak at 350 nm gradually disappeared with the increase in Al^3+^ concentration, and two new absorption peaks appeared at 420 and 303 nm, respectively. The absorption intensity at 420 and 303 nm constantly increased with the increase in Al^3+^ concentration, accompanied with no position change of the peak. The absorbance ratio (y) of probe S at 420 and 303 nm has a good linear relationship with the concentration of Al^3+^ in the range of 1.0 × 10^−6^~6.0 × 10^−5^ mol/L with a detection limit of 5.26 × 10^−8^ mol/L, as shown in Figure 4d. Hence, a ratio absorption method can be established to determine the content of Al^3+^, with high sensitivity and strong anti-interference ability based on the analysis of UV absorption spectra.

### 2.3. Complexation Mechanism of Probes L and S with Al^3+^

The combination mode between L/S with Al^3+^ ions was investigated by utilizing Job’s plot, which was obtained by plotting the molar fraction vs. the changes in the emission intensity at 430 (L) and 420 nm (S), respectively. As shown in Figure 5, the fluorescent intensity reached the maximum value at the mole fraction of about 0.67 (Al^3+^/Al^3+^ + L/S), suggesting the formation of 2:1 stoichiometric complexations between Al^3+^ and L/S.

Moreover, the complexation mechanism and plausible explanation of changes in absorption and fluorescence intensity are illustrated in Figure 2. As mentioned above, fluorescence intensities of L and S gradually increased with increasing Al^3+^ concentration, and UV absorption peak of probe L/S at 345/350 nm gradually disappeared with the increase in Al^3+^concentration, and two new absorption peaks appeared at 430/420 and 302/303 nm, respectively. This phenomenon can be explained as follows. The conjugated double bonds (-C=N-) in Schiff base fluorescent probes can obtain structural isomerization, which would accelerate the non-radiative transition of the Schiff base in the excited state, further leading to a weak fluorescence of the Schiff base. After adding Al^3+^ ions, the metal ions would complex with the Schiff base probes, and the protons of the donor will be removed because of the prevalence of the species optimal for complex formation, preventing structural isomerization of conjugated double bonds (-C=N-), inhibiting the proton transfer process in the excited state of the molecules and resulting in changes of its color and fluorescence behavior [42,43,44]. Simultaneously, previous investigations have also certified that non-bonded electrons of N atoms from C=N took part in coordination with the Al^3+^ ion to inhibit the isomerization process. The coordination of L/S with Al^3+^ ions hindered the rotation around the C=N bond and prevented C=N isomerization, resulting in absorption and fluorescence enhancement [25,41]. Furthermore, there are differences in both fluorescence intensities and UV–Vis absorption of the two probes, which can be explained by the variation in the molecular structure. From the molecular structure, it can be observed that there is a methoxy on the phenyl ring in the L molecule, while no methoxy exists in the S molecule. The existence of methoxy in the L probe would lead to a decrease in both the planarity of the benzene ring and the degree of conjugation of the system, in comparison to that of probe S with no methoxy, further leading to an increase in system energy. This phenomenon resulted in a blue-shift of the emission wavelength from 518 to 516 nm and enhancement of fluorescence intensities and UV–Vis absorption.

### 2.4. Reversibility Experiments

The selective recognition of metal ions by probes mainly depends on the binding dynamics between metal ions and N, O atoms with lone pair electrons in the probe molecules. Herein, reversibility experiments were conducted to study the circulation of the probes. The reversibility of the recognition process of probes L/S was performed by adding an Al^3+^ bonding agent—EDTA. As shown in Figure 6, fluorescence of L/S itself was very weak before the addition of Al^3+^ ions. However, the fluorescence of L/S was greatly enhanced with the addition of Al^3+^ ions, which can be ascribed to the coordination between L/S and Al^3+^ ions, thus emitting bright yellow-green fluorescence. Subsequently, EDTA was added to the system, which resulted in a diminution of the fluorescence intensity at 516 and 518 nm for probes L and S, respectively, indicating regeneration of the free probe L/S. This phenomenon can be attributed to the stronger coordination reaction between Al^3+^ and EDTA, destroying the complexation between Al^3+^ and L/S. All these results could clearly indicate that both probes L and S have good reversibility in the detection of Al^3+^ ions, which is also important for the fabrication of devices to sense the Al^3+^ ions.

### 2.5. Filter Paper Strip Experiments

Filter paper strip experiments were performed to establish another potential application of the probes. Firstly, some thin-layer chromatography (TLC) plates were prepared and coated with probe L/S solution (1 mmol/L) and then dried in air. The prepared TLC strips interacted with various concentrations of Al^3+^, i.e., 0, 0.1, 0.2, 0.3, 0.5 and 1.0 mmol/L. For probe L/S, the color of the strips was observed to change from ginger/yellow to bright yellow-green under a UV chamber by the naked eye, illustrated in Figure 7. These observations clearly indicate that the probe L/S immobilized test strips can also be used for monitoring Al^3+^ in a simple and effective way.

## 3. Experimental Procedure

### 3.1. Reagents and Apparatus

o-Vanillin was purchased from Thain Chemical Technology Shanghai Co., Ltd. (Shanghai, China). Salicylaldehyde, o-aminophenol and p-aminoacetophenone were purchased from Shanghai McLean Biochemical Technology Co., Ltd. (Shanghai, China). Dimethyl sulfoxide was purchased from Tianjin Damao Chemical Reagent Factory (Tianjin, China). All reagents and solvents used in this study were of AR grade. All fluorescence spectra were performed on the F-380 fluorescence spectrometer (Tianjin Gangdong Technology Co., Ltd., Tianjin, China). FT-IR spectra were obtained on avater-370 Fourier infrared spectrometer using KBr plates (American NICO-LET Company, Madison, WI, USA). The ^1^H NMR and ^13^C NMR spectra were tested on a Bruker 400 MHz NMR instrument with TMS as an internal standard (Romanshorn, Switzerland). Ultraviolet spectra were obtained using TU-1901 Double beam UV–Visible spectrophotometer (Beijing Puxi General Instrument Co., Ltd., Beijing, China). Melting points were obtained on WRX-4 micro melting point instrument (Shanghai Yimenshan Instrument and Equipment Co., Ltd., Shanghai, China).

### 3.2. Synthesis

#### 3.2.1. Synthesis of o-Vanillin-p-aminoacetophenone Schiff Base (L)

Firstly, p-aminoacetophenone (0.2703 g, 2 mmol) was dissolved in 25 mL ethanol (EtOH), which were stirred until soluble. Secondly, o-vanillin (0.3043 g, 2 mmol) was added into the above solution, and the mixture was refluxed for 8 h at 80 °C with continuous stirring, which then stewed for 24 h. The resulting reaction suspension was filtrated and washed by EtOH to obtain 0.4217 g (78.3%) of orange product (o-vanillin-p-aminoacetophenone Schiff base, m.p.: 118.2~119.1 °C). RMM: 270.1125 (Appendix A). FT-IR (KBr): 3436 cm^−1^, 1625 cm^−1^, 1537 cm^−1^, 1509 cm^−1^, 1457 cm^−1^, 741 cm^−1^; ^1^H NMR (500 MHz, DMSO-*d*_6_), δ: 12.80 (s, 1H), 8.99 (s, 1H), 8.04 (d, *J* = 8.2 Hz, 2H), 7.53–7.48 (m, 2H), 7.32–7.26 (m, 1H), 7.19–7.13 (m, 1H), 6.93 (t, *J* = 7.9 Hz, 1H), 3.84 (s, 3H), 2.60 (s, 3H), see Appendix A. ^13^C NMR (126 MHz, DMSO-*d*_6_), δ: 197.43, 165.39, 152.58, 151.06, 148.42, 135.38, 130.19, 124.33, 122.03, 119.72, 119.26, 116.44, 56.36, 27.17, Appendix A.

#### 3.2.2. Synthesis of Salicylaldehyde-p-aminoacetophenone Schiff Base (S)

The synthesis of salicylaldehyde-p-aminoacetophenone Schiff base was similar to that of o-vanillin-p-aminoacetophenone Schiff base, except that o-vanillin (0.3043 g, 2 mmol) was substituted for salicylaldehyde (0.2442 g, 2 mmol). The resulting reaction suspension was filtered and washed with EtOH to obtain 0.3474 g (72.6%) of golden yellow product (salicylaldehyde-p-aminoacetophenone Schiff base, m.p.: 112.4~113.2 °C). RMM: 240.1019 (Appendix A). FT-IR(KBr): 3420 cm^−1^, 1630 cm^−1^, 1592 cm^−1^, 1529 cm^−1^, 1487 cm^−1^, 1460 cm^−1^, 1411 cm^−1^; ^1^H NMR (500 MHz, DMSO-*d*_6_), δ: 12.73 (d, *J* = 3.0 Hz, 1H), 8.97 (d, *J* = 3.3 Hz, 1H), 8.02 (dd, *J* = 8.1, 2.5 Hz, 2H), 7.69 (d, *J* = 7.7 Hz, 1H), 7.51–7.41 (m, 3H), 6.99 (dd, *J* = 8.4, 4.8 Hz, 2H), 2.58 (s, 3H), see Appendix A. ^13^C NMR (126 MHz, DMSO-*d*_6_) δ: 197.36, 197.34, 165.22, 160.83, 152.72, 152.70, 135.35, 134.29, 133.11, 130.16, 122.02, 119.76, 119.73, 117.16, 27.13, 27.11, see Appendix A. The synthesized procedure is illustrated in Figure 1.

### 3.3. Spectrophotometric Experiments

An analytical solution of the probe L/S was prepared as 5.0 × 10^−5^ mol/L in 100 mL solution of DMSO/H_2_O (*v*/*v* = 8:2). The metal nitrate solutions were prepared as 5.0 × 10^−3^ mol/L in 50 mL double distilled water (Al^3+^, Mg^2+^, Pb^2+^, Zn^2+^, Cu^2+^, Mn^2+^, Co^2+^, Cr^3+^, Hg^2+^, Cd^2+^, Ni^2+^, Fe^3+^). Fluorescence and UV experiments were performed by gradually increasing the concentration of targeted metal ions to the probes solution to evaluate sensitivity. Moreover, the excitation wavelength for L and S was 430 and 420 nm, respectively, in the fluorescence measurements. The slit widths of both the excitation and emission were 5.0 nm. All the experiments were performed under room temperature.

## 4. Conclusions

Two new types of Schiff base probes L (o-vanillin-p-aminoacetophenone) and S (salicylaldehyde-p-aminoacetophenone) were successfully prepared by using heating reflux method with its simplicity and high yield. The probe L/S could complex with Al^3+^ ions at a 1:2 ratio in DMSO/H_2_O (*v*:*v* = 8:2) solution, which could identify Al^3+^ by ultraviolet visible spectrophotometry and fluorescence with good selectivity, high sensitivity and good reversibility. The LOD of L and S for Al^3+^ was 1.98 × 10^−8^ and 4.79 × 10^−8^ mol/L, respectively, which was lower than most reported studies. The interaction mechanism between L/S and Al^3+^ was explored by Job’s plot, certifying that the protons of the donor will be removed after its coordination with Al^3+^ ions, thereby preventing the structural isomerization of conjugated double bonds (-C=N-), inhibiting the proton transfer process in the excited state of the molecules and resulting in changes in its color and fluorescence behavior. All of the results clearly indicated that the probes have potential applications for the detection of Al^3+^ ions in the environmental system and provide a new strategy for the design and synthesis of multi-functional sensors.

## Data Availability

The data presented in this research are available on request from the corresponding author.

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
