# Peer review of "Schiff Base Compounds as Fluorescent Probes for the Highly Sensitive and Selective Detection of Al3+ Ions"

_molecules, 2023, doi:10.3390/molecules28073090_

Round 1

Reviewer 1 Report

The manuscript "Schiff base compounds as fluorescent probes for the highlysensitive and selective detection of Al3+ ions " is well written and provides a comprehensive overview of the design and synthesis of two new Schiff base fluorescent probes (L and S) for selectively detecting Al3+ ions in aqueous medium. The authors have provided detailed information on the structural characterization of the compounds, their photochromic and fluorescent behaviors, and the selectivity and sensitivity of the probes towards Al3+ ions. The Job's plot was used to verify the 2:1 stoichiometry between the Al3+ and probes, and the limit of detection (LOD) for Al by L and S were 1.98 × 10-8 mol/L and 4.79 × 10-8 mol/L, respectively. The possible recognition mechanism was also discussed. The authors have provided a thorough discussion of the results and potential applications of the probes for the detection of Al3+ ions in the environmental system. Overall, the manuscript is well written and provides a comprehensive overview of the design and synthesis of the probes.

Minor revisions:
1. Reword the introduction to provide a clearer overview of the study.

2. The authors should provide more information on the possible applications of the probes, such as how they can be used in environmental monitoring.

3. The authors should provide more information on the possible reasons recognition mechanism, such as how the coordination of L/S with Al3+ ions hinders the rotation around the C=N bond and prevents the C=N isomerization. In my view, this could be attributed to the prevalence of rigidly defined norms, as evidenced in the literature [Gamov, G.A., et.al(2019) Journal of Molecular Liquids, 283, pp. 825-833. DOI: 10.1016/j.molliq.2019.03.125 ; (2021) Physics and Chemistry of Liquids, 59 (5), pp. 666-678. DOI: 10.1080/00319104.2020.1774878 ;(2021) Journal of Molecular Liquids, 342,  № 117372, . DOI: 10.1016/j.molliq.2021.117372].Please discuss this point in the text of manuscript.

4. Clarify the reversibility experiments section by providing more information on the coordination reaction

Author Response

The manuscript "Schiff base compounds as fluorescent probes for the highly sensitive and selective detection of Al3+ ions" is well written and provides a comprehensive overview of the design and synthesis of two new Schiff base fluorescent probes (L and S) for selectively detecting Al3+ ions in aqueous medium. The authors have provided detailed information on the structural characterization of the compounds, their photochromic and fluorescent behaviors, and the selectivity and sensitivity of the probes towards Al3+ ions. The Job's plot was used to verify the 2:1 stoichiometry between the Al3+ and probes, and the limit of detection (LOD) for Al by L and S were 1.98 × 10-8 mol/L and 4.79 × 10-8 mol/L, respectively. The possible recognition mechanism was also discussed. The authors have provided a thorough discussion of the results and potential applications of the probes for the detection of Al3+ ions in the environmental system. Overall, the manuscript is well written and provides a comprehensive overview of the design and synthesis of the probes.

Minor revisions:

  1. Reword the introduction to provide a clearer overview of the study.

Reply: Great thanks for your comment about this issue. Based on your comment, we have revised the introduction part. And the revised part has been marked with BLUE words. Thanks again.

  1. The authors should provide more information on the possible applications of the probes, such as how they can be used in environmental monitoring.

Reply: We appreciate your valuable comment about this issue. Since the probes exhibited high selectivity and sensitivity towards Al3+ ions, they can be utilized in the detection of Al3+ in water environments. In detail, the samples can be obtained from the actual water environment. Then, appropriate amount of probes were added into the above the solution and the residual Al3+ ions can be detected by V-Vis absorption and fluorescence spectra mentioned in this study. Moreover, the experiments of detecting Al3+ in actual water environment are in progress. Thanks again.

  1. The authors should provide more information on the possible reasons recognition mechanism, such as how the coordination of L/S with Al3+ ions hinders the rotation around the C=N bond and prevents the C=N isomerization. In my view, this could be attributed to the prevalence of rigidly defined norms, as evidenced in the literature [Gamov, G.A., et.al (2019) Journal of Molecular Liquids, 283, pp. 825-833. DOI: 10.1016/j.molliq.2019.03.125; (2021) Physics and Chemistry of Liquids, 59 (5), pp. 666-678. DOI: 10.1080/00319104.2020.1774878; (2021) Journal of Molecular Liquids, 342, 117372, DOI: 10.1016/j.molliq.2021.117372]. Please discuss this point in the text of manuscript.

Reply: Great thanks for your comment about this issue. After adding Al3+ ions, the metal ions would complex with Schiff base probes and the protons of the donor will be removed because of the prevalence of the species optimal for complex formation, preventing the structural isomerization of conjugated double bonds (-C=N-), inhibiting the proton transfer process in the excited state of the molecules and resulting in changes of its color and fluorescence behavior, eventually. In other words, the descriptions mentioned in this manuscript did not conflict with the reviewer’s opinion (please see Line 191–195 in Page 7 and Marked with Blue words). Furthermore, the above references have been added into this manuscript and marked with Blue words (please see Line 427–435 in Page 12). Thanks again for your valuable comment for improving our manuscript.

  1. Clarify the reversibility experiments section by providing more information on the coordination reaction.

Reply: We appreciate your valuable comment about this issue. We have revised the reversibility experiments section in this manuscript. The reversibility of the recognition process of probes L/S was performed by adding an Al3+ bonding agent−EDTA. As shown in Figure 6, fluorescence of L/S itself was very weak before the addition of Al3+ ions. However, the fluorescence of L/S was greatly enhanced with the addition of Al3+ ions, which can be ascribed to the coordination between L/S and Al3+ ions, thus emitting bright yellow green. Subsequently, EDTA was added to the system, which resulted in a diminution of the fluorescence intensity at 516 nm and 518 nm for probes L and S, respectively, indicating the regeneration of the free probe L/S. And this phenomenon can be attributed to the stronger coordination reaction between Al3+ and EDTA, destroying the complexation between Al3+ and L/S. And all these results could clearly indicate that both probes L and S have good reversibility in the detection of Al3+ ions, which is also important for the fabrication of devices to sense the Al3+ ions (Please see Line 215–226 in Page 8 and marked with Blue words). Thanks again.

Reviewer 2 Report

In this paper, the authors introduced two new Schiff bases named as L and S [o-vanillin-p-aminoacetophenone and salicylaldehyde -p-aminoacetophenone] for selective detection of Al3+ ions in an aqueous medium. As well as they proved by the recognition mechanism and its photophysical properties. The described studies could be an interesting extension of the current knowledge. The results should be explained in detail with the supporting experiments. However, in my opinion, the manuscript is not suitable in its current form, and this may be publishable after some issues that the authors should consider before its publication.

Comments

  1. Scheme 1 should be rearranged as they are not logically presented and include all in details of reaction conditions after introduction part (RD section).

2.     Authors should explore the coordination sites of the probes (L and S) with 1H NMR titration and Mass experiments.

3.     Filter paper strip experiments of both probes L/S should be done for further visual identification of the results.

4.     In SI file NMR integration should be mentioned for the better identification of the probe also HRMS/LCMS details must be provided for the newly designed scaffolds. I can see that there are minor trace of impurities in the provided spectrums.

5.     From lines 187-192, the author mentioned that responses of changes with molecular structures of the probes should be explained in detail, in OMe = 516 nm but without OMe = 518 nm.

6.     Several typos have been observed, for ex.; in the abstract, itself line 14 V-Vis must be UV-Vis, in line 194 is it only L to Al+3???, Also, in line 152 mentioned figure 4d but in the titles I did not see 4d/same in figure 3 as well, no need to mention figure s2/s1 in lines 235-237 still mentioned in 278-281, please recheck these types of errors carefully in the whole manuscript. 

Author Response

In this paper, the authors introduced two new Schiff bases named as L and S [o-vanillin-p-aminoacetophenone and salicylaldehyde -p-aminoacetophenone] for selective detection of Al3+ ions in an aqueous medium. As well as they proved by the recognition mechanism and its photo-physical properties. The described studies could be an interesting extension of the current knowledge. The results should be explained in detail with the supporting experiments. However, in my opinion, the manuscript is not suitable in its current form, and this may be publishable after some issues that the authors should consider before its publication.

  1. Scheme 1 should be rearranged as they are not logically presented and include all in details of reaction conditions after introduction part (RD section).

Reply: Great thanks for your comment about this issue. Based on your comment, we have revised and rearranged the two schemes (Scheme 1: Synthesized procedure of probes L and S; Scheme 2: Possible recognition mechanism of probe L/S to Al3+ ions). (Please see Line 83–85 in Page 2 and Line 209 in Page 8, marked with Blue words in this manuscript). Thanks again.

  1. Authors should explore the coordination sites of the probes (L and S) with 1H NMR titration and Mass experiments.

Reply: We appreciate your valuable comment about this issue. Definitely, 1H NMR titration and Mass experiments will contribute to analyzing the coordination sites of the probes (L and S). However, the above two experiments can’t be performed according to our current laboratory conditions. And we will dedicate ourselves to solving these problems in the following stage. Thanks again.

  1. Filter paper strip experiments of both probes L/S should be done for further visual identification of the results.

Reply: We appreciate your valuable comment about this issue. Based on your comment, we have added the filter paper strip experiments into our manuscript. Firstly, some thin-layer chromatography (TLC) plates were prepared and coated with probe L/S solution (1 mmol/L) and then dried in air. And the prepared TLC strips were interacted with various concentrations of Al3+, i.e. 0, 0.1, 0.2, 0.3, 0.5 and 1.0 mmol/L. For probe L/S, the color of the strips was observed to change from ginger/yellow to bright yellow green under a UV chamber by the naked eye, illustrated in Figure 7. These observations clearly indicated that the probe L/S immobilize test strips can also be used for monitoring Al3+ in a simple and effective way. Please see Line 230–234 in Page 8 and Line 235–242 in Page 9, marked with Blue words in this manuscript. Thanks again.

Figure 7 Fluorescence color changes of test strips for Al3+ at different concentrations (0, 0.1, 0.2, 0.3, 0.5 and 1.0 mmol/L) under a UV lamp.

  1. In SI file NMR integration should be mentioned for the better identification of the probe also HRMS/LCMS details must be provided for the newly designed scaffolds. I can see that there are minor traces of impurities in the provided spectra.

Reply: Thanks for your comment about this issue. 1H NMR and 13C NMR integrations of the two probes have been added into the spectra Moreover, based on your comment, we have added the HRMS of probes L and S into Supplementary Materials, see Figure S1 and S4, respectively. From HRMS, it can be seen that the relative molecular mass (RMM) of probe L and S is 270.1125 and 240.1019, respectively, which is consistent with their theoretical RMM values (269.2952 for probe L and 239.2693 for probe S), indicating the high purities of the obtained probes. Thanks again.

  1. From lines 187-192, the author mentioned that responses of changes with molecular structures of the probes should be explained in detail, in OMe = 516 nm but without OMe = 518 nm.

Reply: We appreciate your valuable comment about this issue. The existence of methoxy in L probe would lead to the decrease in both the planarity of the benzene ring and the degree of conjugation of the system, in comparison to that of probe S with no methoxy, further leading to the increase of system energy. And this phenomenon resulted in the blue-shift of emission wavelength from 518 nm to 516 nm and enhancement of fluorescence intensities and UV-Vis absorption, eventually. The corresponding descriptions has been added into this manuscript and Marked with Blue words. Please see Line 203–208 in Page 8. Thanks again.

  1. Several typos have been observed, for ex.; in the abstract, itself line 14 V-Vis must be UV-Vis, in line 194 is it only L to Al3+???, Also, in line 152 mentioned figure 4d but in the titles I did not see 4d/same in figure 3 as well, no need to mention figure s2/s1 in lines 235-237 still mentioned in 278-281, please recheck these types of errors carefully in the whole manuscript.

Reply: Thanks for your comment about this issue. Based on your comment, we have checked these types of errors carefully in the whole manuscript. Figure S1 and S2 mentioned in this manuscript were based on the format requirements of Journal of Molecules. Furthermore, the whole manuscript has been revised carefully and marked with Blue words. Thanks again.

Round 2

Reviewer 2 Report

The authors improved the suggested changes accordingly and can be accepted for publication.